# Do Not Leave Your Patients in the Dark—Using American College of Rheumatology and European Alliance of Associations for Rheumatology Recommendations for Vaccination in Polish Adult Patients with Autoimmune Inflammatory Rheumatic Diseases

**DOI:** 10.3390/vaccines11121854

**Published:** 2023-12-14

**Authors:** Jakub Wroński, Karolina Palej, Sandra Stańczyk, Marta Łosoś, Joanna Werońska-Tatara, Małgorzata Stasiek, Marta Wysmołek, Agnieszka Olech, Anna Felis-Giemza

**Affiliations:** 1Department of Rheumatology, National Institute of Geriatrics, Rheumatology and Rehabilitation, Spartańska 1, 02-637 Warsaw, Poland; agnieszka.olech@spartanska.pl; 2Biologic Therapy Center, National Institute of Geriatrics, Rheumatology and Rehabilitation, 02-637 Warsaw, Poland; karolina.palej89@gmail.com (K.P.); stanczyk.sandra@gmail.com (S.S.); los.mar@wp.pl (M.Ł.); asiaweronska@wp.pl (J.W.-T.); margo1801@o2.pl (M.S.); marta.wysmolek@gmail.com (M.W.); annafelis@wp.pl (A.F.-G.)

**Keywords:** autoimmune inflammatory rheumatic diseases, coverage, uptake, vaccination, vaccine

## Abstract

(1) Introduction: Patients with autoimmune inflammatory rheumatic diseases (AIIRD) face a higher infectious risk compared to the general population. As per the ACR and EULAR recommendations, vaccinations against influenza, COVID-19, pneumococci, and tetanus are recommended for most patients with AIIRD. (2) Objectives: This study aimed to assess vaccination coverage among Polish AIIRD patients and identify factors influencing it. (3) Patients and Methods: This study was conducted at the reference rheumatological center in Poland between May 2023 and October 2023. The study participants completed a questionnaire covering their knowledge of vaccination recommendations, actual vaccination status, factors affecting their decision to vaccinate, and their perspectives on immunization. (4) Results: This study involved 300 AIIRD patients and 60 controls. Both groups exhibited comparably low vaccination rates for all diseases (the highest for COVID-19—52% in both groups and the lowest for pneumococci—7.7% and 10%, respectively). Knowledge about recommended vaccinations was limited among patients in both groups. AIIRD patients were also not aware that they should avoid live vaccines. The primary motivators for vaccination among AIIRD patients were fear of infection (up to 75%) and medical advice (up to 74.6%). Conversely, the predominant reasons for non-vaccination were a lack of knowledge that vaccination is recommended (up to 74.7%) and concerns about potential adverse effects (up to 48.6%). Many patients reported not receiving vaccination recommendations from either primary care physicians or rheumatologists. (5) Conclusions: To enhance vaccination coverage among AIIRD patients in Poland, it is essential to educate them about vaccinations during routine medical consultations, emphasizing the increased risk of infection, informing them about recommended vaccinations, and clarifying doubts about adverse effects.

## 1. Introduction

The COVID-19 pandemic has reminded the medical community of the importance of vaccinations in the prevention of infectious diseases. Protective vaccinations (not only against COVID-19) are even more important in immunocompromised patients, which include patients with autoimmune inflammatory rheumatic diseases (AIIRD). Patients with AIIRD, due to immunological disorders resulting from the underlying disease itself and the immunomodulatory drugs taken, are at greater risk of infectious diseases compared to the general population [1,2]. However, the use of vaccines in patients treated with immunomodulatory drugs may be associated with limited immunogenicity of vaccinations, especially in patients treated with rituximab, mycophenolate mofetil, abatacept, and glucocorticosteroids [3].

According to the current recommendations of international rheumatological scientific societies—American College of Rheumatology (ACR) 2022 [1] and European Alliance of Associations for Rheumatology (EULAR) 2019 [2], as well as detailed guidelines for vaccination against COVID-19 (ACR 2022 [4], EULAR 2021 [5])—in the majority of AIIRD patients, the following are recommended: annual influenza vaccinations, COVID-19 vaccinations including booster doses, and pneumococcal vaccinations. As per EULAR recommendation, tetanus and HPV vaccinations should be recommended as mentioned in the general population guidelines, that is, tetanus vaccine every 10 years and HPV vaccine in adult patients between 18 and 26 years of age. However, as per ACR recommendations, HPV vaccination may also be considered for AIIRD patients taking immunomodulatory drugs before the age of 45. ACR also recommends herpes zoster vaccinations for most AIIRD patients. In the EULAR recommendations, vaccination against herpes zoster was only recommended in high-risk groups, but these guidelines were created before data on the effectiveness of the new recombinant vaccine against herpes zoster were available.

There is a lack of epidemiological data on the vaccination status of AIIRD patients in Poland. According to a small study of 57 patients with AIIRD, vaccination coverage in this group may be higher than in the general population—78.9% of AIIRD patients were vaccinated against COVID-19 (primary schedule) and 31% against influenza [6]. However, apart from the fact that this study included a very small group of patients, it was conducted during the COVID-19 pandemic. The COVID-19 pandemic, probably through increased fear of infectious diseases and pro-vaccination recommendations, resulted in increased uptake of influenza vaccination in AIIRD patients [7]. Data on other recommended vaccines for AIIRD patients are unavailable in Poland.

Our study aimed to assess vaccination coverage in accordance with the ACR and EULAR recommendations in a large group of Polish AIIRD patients. We selected vaccinations recommended for most AIIRD patients—vaccination against influenza, COVID-19, pneumococci, and tetanus. This study did not include vaccination against herpes zoster (the first herpes zoster recombinant vaccine became available to patients in Poland only in the second quarter of 2023) and against HPV (as there is no recommendation for HPV vaccination in most people over 26 years of age). Additionally, this study aimed to determine factors that may influence the level of vaccination coverage.

## 2. Patients and Methods

### 2.1. Patients

This study was conducted at the Department of Rheumatology and Biologic Therapy Center in the National Institute of Geriatrics, Rheumatology, and Rehabilitation in Warsaw, Poland, between May 2023 and October 2023. The study participants were divided into two groups: the study group (AIIRD group) and the control group. The inclusion criteria for the AIIRD group were age ≥ 18 and the diagnosis of AIIRD before admission to the hospital. For the control group, inclusion criteria were age ≥ 18 and no previous diagnosis of AIIRD, with the exclusion criterion being current immunosuppressive treatment (the group consisted of patients hospitalized in the rheumatology department referred for diagnostics). The study protocol was approved by the hospital bioethics committee (no. KBT-3/8/2023). All participants signed informed consent for inclusion in this study. This study was conducted according to the Declaration of Helsinki.

### 2.2. Methods

The study participants received a questionnaire (available in the Polish language in Appendix A) with questions regarding knowledge about recommended vaccinations, vaccination coverage, determinants influencing their decision to undergo a particular vaccination, and their perspectives on immunization. Additionally, demographic data (regarding age, gender, education level, place of residence, employment, wealth, and relationship status) were collected based on a questionnaire. In the AIIRD group, data on the type of disease, disease duration, and treatment received (including glucocorticosteroids (GCs) and disease-modifying antirheumatic drugs (DMARDs)) were collected based on patients’ electronic medical records.

The state of knowledge and vaccination recommendation utilization were compared with the currently applicable recommendations—in the case of the AIIRD group, the ACR and EULAR recommendations, and in the case of the control group, the national “Preventive Vaccination program for 2023” [8] and “Announcement of the Minister of Health on the implementation of booster vaccinations against COVID-19 in the National Vaccination Program” [9]. The Polish Ministry of Health recommends every adult be vaccinated with at least two booster doses of the COVID-19 vaccine, annually against influenza, and once every 10 years against tetanus. Pneumococcal vaccination is recommended for people aged ≥ 50.

### 2.3. Statistics

The compliance of the data with the normal distribution was assessed using the Shapiro–Wilk test. The significance of the observed differences between the groups was assessed using the Student’s T test for variables with a normal distribution, the Mann–Whitney U test for variables without a normal distribution, and for categorical variables, the Chi-squared test or Fisher’s exact test for values less than 5. Logistic regression and an odds ratio (OR) with a 95% CI were used to identify predictive factors associated with vaccination. The final multivariate model was created using the stepwise backward method; variables from the univariate analysis with a likelihood-ratio *p*-value less than 0.1 were used. Statistical analysis was performed using Statistica 13.3 software (StatSoft Polska, Cracow, Poland).

## 3. Results

This study involved 360 participants—300 in the AIIRD group and 60 in the control group. The demographics of participants in both groups did not significantly differ (Table 1). The majority of participants were women (68% in AIIRD and 63.3% in the control group), with a median age of 49 and 52 years (in AIIRD and control groups, respectively). AIIRD patient characteristics are shown in Table 2. Most patients had arthritis—various types of spondyloarthritis (44%), and rheumatoid arthritis (35.7%). Most patients (66.3%) were also treated with biological DMARDs (bDMARDs) or targeted synthetic DMARDs (tsDMARDs).

Vaccination coverage for all types of vaccinations did not significantly differ between the two groups (Figure 1). Most participants were vaccinated against COVID-19 (52% in both groups with at least two booster doses) and the least against pneumococci (7.7% and 10%—in the AIIRD and control groups, respectively).

Both groups not only did not get vaccinated following the recommendations, but most of them did not know the current recommendations. Similar to the vaccination coverage, most participants correctly recognized the need to be vaccinated against COVID-19 (63.3% in the AIIRD group and 50% in the control group), and the fewest were aware of the need to be vaccinated against pneumococci (17% in both groups). The differences between the groups correctly identifying the need to get vaccinated remained statistically insignificant, although the AIIRD group was statistically less likely to admit not knowing the recommendations (Table 3). Only 25.7% of patients treated with immunomodulatory drugs knew that they should avoid live vaccines.

The primary factors motivating vaccination among AIIRD patients included recommendations from the attending physician (up to 74.6%) and the apprehension of contracting illness (up to 75%). Conversely, the principal reasons for non-vaccination among AIIRD patients encompassed a lack of knowledge that vaccination is recommended (up to 74.7%), concerns about potential adverse effects (up to 48.6%), and exacerbation of rheumatic disease (up to 37.8%). A detailed breakdown of determinants influencing patients’ decisions for or against a particular vaccination is shown in Figure 2. This study also assessed opinions on the effectiveness of vaccinations, their safety, and the possibility of exacerbation of rheumatic disease. Most AIIRD patients stated that they lack the knowledge to form opinions (58.7–63.7%). Detailed opinion data are provided in Table 4.

In the pursuit of elucidating the factors contributing to the diminished vaccination rate, participants were additionally queried about information regarding vaccination recommendations that they received from their general practitioner and (in the case of the AIIRD group) their rheumatologist (Table 5). In both groups, a comparable number of patients reported receiving a recommendation to be vaccinated from their family doctor—the fewest against pneumococci (3.7% AIIRD and 5.8% control), the most against COVID-19 (48.3% and 38.5%, respectively). AIIRD patients were less likely to report receiving vaccination recommendations from a rheumatologist than from a family doctor against COVID-19 (*p* < 0.001) and tetanus (*p* = 0.03), but comparably often against influenza and pneumococci. The patients under constant care from the same doctor more often received recommendations to vaccinate against COVID-19 (*p* = 0.047, same family doctor; *p* = 0.044, same rheumatologist), and in the AIIRD group also against tetanus (*p* = 0.047, same rheumatologist).

In the multivariate analysis, some demographic characteristics influenced the level of patient vaccination for certain vaccines. People living in cities with over 500,000 inhabitants (OR = 3.78, 95% CI = 1.67–8.52), being in a relationship (OR = 2, 95% CI = 1.01–3.95), and older people (OR = 1.03, 95% CI = 1.01–1.05) were more likely to be vaccinated against COVID-19. On the other hand, older people were less likely to be vaccinated against tetanus (OR = 0.96, 95% CI = 0.93–0.99). Among AIIRD patients, the type of immunomodulatory treatment (including bDMARDs/tsDMARDs vs. cDMARDs) did not affect vaccination status.

## 4. Discussion

Unfortunately, despite the important role of vaccinations and the detailed guidelines available, Poland is characterized by a very low level of vaccination among its population. Only 59.6% of Poles have been vaccinated with the COVID-19 primary schedule since the beginning of the pandemic (despite forced restrictions for unvaccinated people until April 2022) [10]. In the 2022/2023 season, only 5.65% of the population was vaccinated against influenza [11]. According to the latest ECDC report (data from the 2017–2018 season), Poland has the third lowest uptake of influenza vaccination among European Union countries [12]. The low vaccination rate among AIIRD patients found in our study is not surprising in this context but fits into the available literature data. The vaccination rate among AIIRD patients varies between different countries and is between 1.5 and 92.4% for influenza vaccines [6,7,13,14,15,16,17,18,19,20,21,22,23,24,25,26,27,28,29,30,31,32,33,34,35,36,37,38,39,40,41,42,43], 5.8 and 71.2% for the pneumococcal vaccine [13,15,16,17,18,20,21,22,23,24,25,26,27,28,29,30,31,32,33,34,37,38,41,42,43,44,45,46], 24.3 and 83.7% for tetanus vaccination (booster within 10 years) [26,27,29,33,41,42], and 35.8 and 98.1% for the COVID-19 vaccine (following the recommendations applicable at the time of this study) [6,40,41,42,43,47,48,49]. An additional problem is the fact that vaccination coverage in Polish AIIRD patients is comparably low as in the general Polish population, even though AIIRD patients constitute a group of particular infectious risks. However, it should not be surprising that patients do not get vaccinated if they do not know what they should be vaccinated against. Alas, our study showed a very low level of knowledge among AIIRD patients regarding specific vaccination recommendations. Importantly, the lack of knowledge also concerned the need to avoid live vaccines by people using immunomodulatory drugs, which may pose a direct threat to patients.

The aim of this study was not only to find out the exact vaccination coverage in Polish AIIRD patients but, equally importantly, the reasons for this condition and what can be implemented to improve vaccination rates. This study clearly showed that the main motivators for vaccination among AIIRD patients were the fear of contracting the disease and receiving a vaccination recommendation from the attending physician. Disturbingly, most patients reported that they did not receive such recommendations—neither from their general practitioner nor (even less often) from a rheumatologist. Certainly, this does not imply that they did not receive such recommendations, but it may mean that recommendations have been ineffectively communicated to them. All patients starting bDMARDs/tsDMARDs in our institute are informed to avoid live vaccines—and still, only 14.8% of them reported being instructed to do so. This is even though these patients must read and sign the “*Regulations of the Biologic Therapy Center*”, which include, among others, instructions to avoid live vaccines, before starting therapy. Even though patients had no recollection of being informed by their rheumatologist, written instructions allowed a statistically significant increase in the percentage of patients aware that they should avoid live vaccines—from 19.1% in patients treated with cDMARDs to 31.1% in patients treated with bDMARDs/tsDMARDs (*p* = 0.02). Notably, this study revealed which factors do not affect the level of vaccination—e.g., implementing restrictions on individuals unvaccinated against COVID-19 proves to be ineffective. A comparable proportion of respondents indicated that such restrictions served as a motivation for vaccination (15.8%) and a deterrent against vaccination (12.1%). Among the main reasons against getting vaccinated, AIIRD patients mentioned, in addition to the lack of knowledge that vaccination is recommended, concerns about potential adverse effects, the exacerbation of rheumatic disease, and lack of concerns regarding contracting illness. This is in line with the literature data that shows the factor mainly affecting vaccination uptake is lack of recommendation by physicians, followed by lack of knowledge that vaccination is required and the fear of their side effects [7,15,18,19,20,22,23,24,25,28,29,30,31,32,33,34,40,50].

This indicates the most important topics that should be discussed by a doctor when talking about vaccinations with AIIRD patients: (1) Increased risk of infectious diseases in AIIRD patients; (2) Recommendations for vaccination with specific vaccines; (3) Ensuring the safety of vaccination in AIIRD patients and answering patients’ doubts; (4) Informing patients that the risk of exacerbations of rheumatic disease after vaccination is small, that exacerbations are temporary, and that they do not require a change in therapy. All tasks should be performed by rheumatologists during routine visits. However, in the Polish health care system, vaccinations take place at the level of primary health care clinics. It is therefore important for the rheumatologist to (5) prepare appropriate AIIRD patient vaccination guidelines for primary care physicians so that they can order appropriate vaccinations. As our research shows, informing the patient once is not enough to obtain an adequate level of knowledge. Finally, (6) all steps listed in points 1–5 should be repeated, e.g., every year before starting seasonal flu vaccinations. A summary of our recommendations to improve AIIRD patients’ knowledge about vaccinations is presented in Table 6. Of course, one needs to be aware of the barriers that may prevent rheumatologists from following these recommendations in their everyday clinical practice. The only available study shows that rheumatologists’ infrequent recommendation of vaccinations in AIIRD patients may result from insufficient time during the visit, forgetfulness, and unfamiliarity with guidelines [44]. An international survey of 371 rheumatologists showed that only 46.7% of them rated their familiarity with the 2019 EULAR guideline as “well” or “very well” [51].

In addition to improving patient information, various systemic solutions are also possible. Active vaccination campaigns at the rheumatologic outpatient clinic (vaccinations during routine rheumatology visits), reminders to staff and physicians, assigning key interventions to nonphysician personnel and educating them about disease and vaccine guidelines, physician auditing and feedback, and patient outreach have shown effectiveness in increasing vaccination rates in AIIRD patients [14,16,36,39,45]. Though often low-cost and simple, all of these solutions require decisions at the system level. The most important thing is that, as a final result, the patient should be offered an appropriate vaccination, as some studies show nearly all AIIRD patients agree to be vaccinated when offered [23,25,42].

Our study is the first to assess the utilization of ACR and EULAR recommendations for vaccination in Polish adult patients with autoimmune inflammatory rheumatic diseases. In addition, our study evaluated the vaccination coverage of all available in-local-settings-recommended vaccines in AIIRD patients, not only influenza and pneumococcal vaccination. The strength of our study is also the inclusion of the control group and the analysis of multiple factors influencing low vaccination coverage. However, this study had several limitations. The nature of this survey study design means that it was prone to recall bias. Vaccination coverage was only self-reported. In addition, there was likely selection bias—some patients refused to complete the survey. These could be patients who are skeptical about vaccinations, so the vaccination uptake found in this study may be overestimated. This study was also conducted in a reference center in the capital city, and one should be cautious about the generalization of its results to the rest of Poland. Still, more than half of the patients included in this study were people from rural areas and small towns, and in the multivariate analysis, the place of residence only influenced the level of vaccination against COVID-19. Despite the above limitations, this study provided valuable data that allowed the formulation of recommendations on how to improve the vaccination rate of AIIRD patients in Poland.

## 5. Conclusions

Our study showed that we are keeping patients in the dark. Patients’ ignorance directly translates into low vaccination rates. It is the role of rheumatologists and family doctors to change this state of affairs. Although systemic solutions would facilitate the improvement of vaccination coverage among AIIRD patients, one should first start with an examination of conscience—have I informed my patients about the recommended vaccinations?

## Figures and Tables

**Figure 1 vaccines-11-01854-f001:**
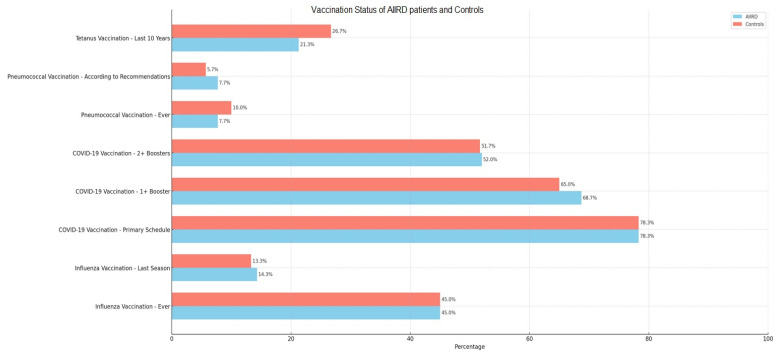
Vaccination status of autoimmune inflammatory rheumatic disease patients and controls. According to Polish recommendations for the general adult population, every adult should be vaccinated with seasonal influenza vaccination, at least two booster doses of COVID-19 vaccine, and once every 10 years against tetanus. Pneumococcal vaccination is recommended for those aged ≥ 50.

**Figure 2 vaccines-11-01854-f002:**
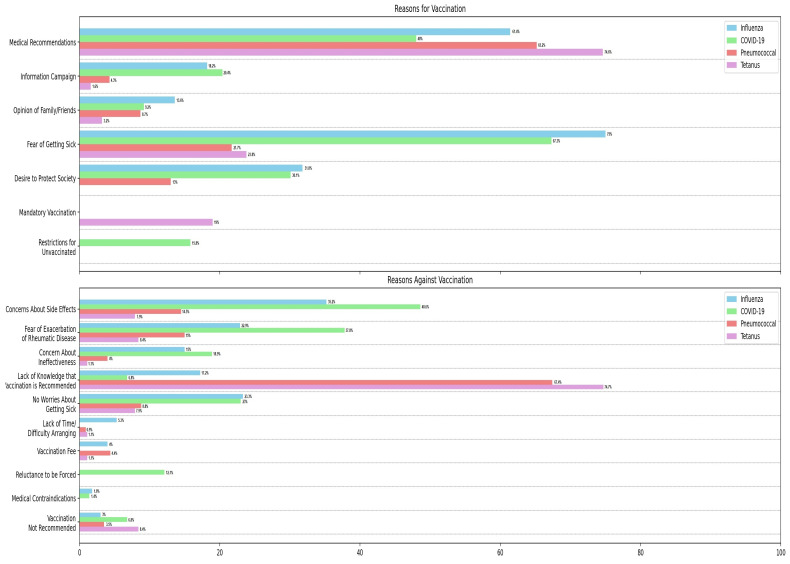
Reasons for and against getting vaccinated reported by autoimmune inflammatory rheumatic disease patients. AIIRD—autoimmune inflammatory rheumatic diseases.

**Table 1 vaccines-11-01854-t001:** Participants’ demographics. AIIRD—autoimmune inflammatory rheumatic diseases.

	AIIRD, n = 300	Controls, n = 60	Difference
Age, median (range)	49 (19–84)	52 (21–87)	*p* = 0.32
Sex, female n (%)	204 (68%)	38 (63.3%)	*p* = 0.48
Education level (n, %)			
- primary	4 (1.3%)	0 (0%)	*p* = 1
- secondary	128 (42.7%)	22 (36.7%)	*p*= 0.39
- higher	142 (47.3%)	29 (48.3%)	*p* = 0.89
- unknown	26 (8.7%)	9 (15%)	*p* = 0.13
Place of residence (n, %)			
- village	60 (20%)	11 (18.3%)	*p* = 0.77
- town up to 100k inhabitants	97 (32.3%)	18 (30%)	*p* = 0.72
- city 100k–500k inhabitants	18 (6%)	6 (10%)	*p* = 0.26
- city over 500k inhabitants	96 (32%)	18 (30%)	*p* = 0.76
- unknown	29 (9.7%)	7 (11.7%)	*p* = 0.64
Employment status (n, %)			
- employed	166 (55.3%)	28 (48.3%)	*p* = 0.22
- pension	98 (32.7%)	19 (31.7%)	*p* = 0.88
- unemployed	10 (3.3%)	3 (5%)	*p* = 0.46
- student	2 (0.7%)	2 (3.3%)	*p* = 0.13
- unknown	24 (8%)	7 (11.7%)	*p* = 0.35
Wealth status (n, %)			
- can afford some luxury	20 (6.7%)	5 (8.3%)	*p* = 0.59
- have enough for a lot without saving	123 (41%)	23 (38.3%)	*p* = 0.7
- have enough on a daily basis; have to save for serious purchases	113 (37.7%)	21 (35%)	*p* = 0.7
- have to be very frugal on a daily basis	17 (5.7%)	6 (10%)	*p* = 0.21
- don’t have enough for basic needs	1 (0.3%)	0 (0%)	*p* = 1
- unknown	26 (8.7%)	5 (8.3%)	*p* = 1
Relationship status (n, %)			
- single	55 (18.3%)	10 (16.7%)	*p* = 0.76
- in relationship	208 (69.3%)	42 (70%)	*p* = 0.92
- unknown	37 (12.3%)	8 (13.3%)	*p* = 0.83

**Table 2 vaccines-11-01854-t002:** Autoimmune inflammatory rheumatic diseases group characteristics. AIIRD—autoimmune inflammatory rheumatic diseases; bDMARDs—biological disease-modifying antirheumatic drugs; cDMARDs—conventional disease-modifying antirheumatic drugs; tsDMARDs—targeted synthetic disease-modifying antirheumatic drugs.

AIIRD, n = 300	
Disease (n, %)	
- Rheumatoid arthritis	107 (35.7%)
- Ankylosing Spondylitis	72 (24%)
- Psoriatic Arthritis	43 (14.3%)
- Non-Radiographic Spondyloarthritis	17 (5.7%)
- Systemic Sclerosis	10 (3.3%)
- Juvenile Idiopathic Arthritis	9 (3%)
- Sjogren’s Syndrome	9 (3%)
- Vasculitis	8 (2.7%)
- Systemic Lupus Erythematosus	6 (2%)
- Polymyalgia rheumatica	4 (1.3%)
- Still Disease	4 (1.3%)
- Myositis	3 (1%)
- Other	6 (2%)
- Unknown	2 (0.7%)
**Disease Duration (Median, Range)**	
- from first symptoms in months	108 (1–492)
- from diagnosis in months	78 (1–480)
**Glucocorticoids**	
- treatment, n (%)	60 (20%)
- dose in mg/d of prednisone, median (range)	5 (1–20)
- duration in months, median (range)	48 (0–456)
**cDMARDs Treatment (n, %)**	
- methotrexate	118 (39.3%)
- sulfasalazine	23 (7.7%)
- hydroxychloroquine	23 (7.7%)
- mycophenolate mofetil	8 (2.7%)
- leflunomide	7 (2.3%)
- azathioprine	3 (1%)
- cyclophosphamide	3 (1%)
- cyclosporine	1 (0.3%)
**bDMARDs and tsDMARDs (n, %)**	
All	199 (66.3%)
- TNF inhibitors	133 (44.3%)
- IL-6 inhibitors	20 (6.7%)
- IL-17 inhibitors	20 (6.7%)
- JAK inhibitors	16 (5.3%)
- anti CD20	5 (1.7%)
- IL-1 inhibitors	1 (0.3%)
- IL-23 inhibitors	1 (0.3%)
- TYK inhibitors	1 (0.3%)
- unknown	2 (0.7%)
- Combined therapy, n (%)	94 (31.3%)

**Table 3 vaccines-11-01854-t003:** State of knowledge regarding recommended vaccinations in autoimmune inflammatory rheumatic diseases patients and controls. AIIRD—autoimmune inflammatory rheumatic diseases. According to Polish recommendations for the general adult population, every adult should be vaccinated with a seasonal influenza vaccination, at least two booster doses of the COVID-19 vaccine, and once every 10 years against tetanus. Pneumococcal vaccination is recommended for those aged ≥ 50. Statistically significant differences are marked in bold.

Should You Be Vaccinated with:	AIIRD n = 300	Controls n = 60	Difference
Influenza vaccination every season?			
- Yes (recommended)	124 (41.3%)	26 (43.3%)	*p* = 0.77
- No	56 (18.7%)	14 (23.3%)	*p* = 0.4
- Don’t know	111 (37%)	18 (30%)	*p* = 0.3
- Missing data	9 (3%)	2 (3.3%)	*p* = 1
COVID-19 vaccination?			
- Yes (recommended)	190 (63.3%)	30 (50%)	*p* = 0.53
- No	37 (12.3%)	8 (13.3%)	*p* = 0.83
- Don’t know	67 (22.3%)	21 (35%)	***p* = 0.04**
- Missing data	6 (2%)	1 (1.7%)	*p* = 1
COVID-19 booster?			
- Yes, 2 doses (recommended)	146 (48.7%)	24 (40%)	*p* = 0.22
- Yes, 1 dose	17 (5.7%)	2 (3.3%)	*p* = 0.75
- No	43 (14.3%)	10 (16.7%)	*p* = 0.64
- Don’t know	81 (27%)	24 (40%)	***p* = 0.04**
- Missing data	13 (4.3%)	0 (0%)	*p* = 0.14
Pneumococcal vaccination?			
- Yes	51 (17%)	7 (11.7%)	*p* = 0.31
- No	35 (11.7%)	10 (16.7%)	*p* = 0.29
- Don’t know	203 (67.7%)	43 (71.7%)	*p* = 0.54
- Missing data	11 (3.7%)	0 (0%)	*p* = 0.22
- Answered as per recommendations	51 (17%)	10 (16.7%)	*p* = 0.95
Tetanus vaccination every 10 years?			
- Yes (recommended)	62 (20.7%)	6 (10%)	*p* = 0.05
- No	46 (15.3%)	3 (5%)	***p* = 0.03**
- Don’t know	184 (61.3%)	50 (83.3%)	***p* = 0.001**
- Missing data	8 (2.7%)	1 (1.7%)	*p* = 1
Are live vaccines contraindicated to you?			
- Yes (recommended)	80 (25.7%)	-	-
- No	30 (10%)	-	-
- Don’t know	179 (59.7%)	-	-
- Missing data	11 (3.7%)	-	-

**Table 4 vaccines-11-01854-t004:** Opinions regarding vaccinations in autoimmune inflammatory rheumatic diseases patients. AIIRD—autoimmune inflammatory rheumatic diseases.

Do You Think Vaccinations:	AIIRD Group n = 300
Are effective in rheumatological patients?	
- Yes	105 (35%)
- No	13 (4.3%)
- Don’t know	176 (58.7%)
- Missing data	8 (2.7%)
Are safe in rheumatological patients?	
- Yes	96 (32%)
- No	12 (4%)
- Don’t know	182 (60.7%)
- Missing data	10 (3.3%)
May exacerbate rheumatic disease?	
- Yes	48 (16%)
- No	49 (16.3%)
- Don’t know	191 (63.7%)
- Missing data	12 (4%)

**Table 5 vaccines-11-01854-t005:** Vaccination recommendations received from family doctors and rheumatologists. AIIRD—autoimmune inflammatory rheumatic diseases.

	AIIRD n = 300	Controls n = 60	Difference
Is under the care of a family doctor	267 (89%)	52 (86.6%)	*p* = 0.60
Is under constant care of the same doctor	183 (68.5%)	35 (67.3%)	*p* = 0.86
The doctor recommended vaccination against:			
Influenza	59 (22.1%)	9 (17.3%)	*p* = 0.44
COVID-19 primary schedule	129 (48.3%)	20 (38.5%)	*p* = 0.19
COVID-19 with 1 booster dose	14 (5.2%)	1 (1.9%)	*p* = 0.48
COVID-19 with 2 booster doses	92 (34.5%)	16 (30.8%)	*p* = 0.61
Pneumococci	10 (3.7%)	3 (5.8%)	*p* = 0.45
Tetanus	30 (11.2%)	4 (7.7%)	*p* = 0.62
Is under the care of a rheumatologist	280 (93.3%)	-	-
Is under constant care of the same rheumatologist	194 (69.3%)		
The rheumatologist recommended vaccination against:			
Influenza	54 (19.3%)	-	-
COVID-19 primary schedule	94 (33.6%)	-	-
COVID-19 with 1 booster dose	7 (2.5%)	-	-
COVID-19 with 2 booster doses	69 (24.6%)	-	-
Pneumococci	10 (3.6%)	-	-
Tetanus	17 (6.1%)	-	-
The doctor recommended avoiding live vaccines	33 (11.8%)	-	-

**Table 6 vaccines-11-01854-t006:** Summary of recommendations for rheumatologists to improve AIIRD patients’ knowledge about vaccinations.

During Routine Rheumatological Visits, Inform the Patient about:
(1) Increased risk of infectious diseases in AIIRD patients
(2) Recommendations for vaccination with specific vaccines
(3) Ensure the safety of vaccination in AIIRD patients and answer patients’ doubts
(4) Inform that the risk of exacerbation of rheumatic disease after vaccination is small, exacerbations are temporary and do not require a change in therapy.
(5) Prepare appropriate vaccination guidelines for patients’ primary care physicians.
(6) All steps listed in points 1–5 should be repeated, e.g., every year before starting seasonal flu vaccinations

## Data Availability

Available upon reasonable request sent to the corresponding author.

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
