# Peer review of "Do Not Leave Your Patients in the Dark—Using American College of Rheumatology and European Alliance of Associations for Rheumatology Recommendations for Vaccination in Polish Adult Patients with Autoimmune Inflammatory Rheumatic Diseases"

_vaccines, 2023, doi:10.3390/vaccines11121854_

Round 1
Reviewer 1 Report
Comments and Suggestions for Authors
In the Manuscript entitled “Keeping patients in the dark – utilization of ACR and EULAR recommendations for vaccination in Polish adult patients with autoimmune inflammatory rheumatic diseases” the authors provide a study that aims to determine vaccination coverage and factors influencing it in Polish AIIRD patients. It is exciting work due to the importance of vaccination in preventing several diseases and reducing financial losses to health systems. The work is good, but I believe the authors must provide figures in graphic format instead of only tables for the readers to understand them better.
Comments on the Quality of English Languagethe English language must be improved
Author Response
- In the Manuscript entitled “Keeping patients in the dark – utilization of ACR and EULAR recommendations for vaccination in Polish adult patients with autoimmune inflammatory rheumatic diseases” the authors provide a study that aims to determine vaccination coverage and factors influencing it in Polish AIIRD patients. It is exciting work due to the importance of vaccination in preventing several diseases and reducing financial losses to health systems. The work is good, but I believe the authors must provide figures in graphic format instead of only tables for the readers to understand them better.
Thank you for your kind review of our work. As suggested, we have replaced the data from tables 3 and 5 with figures 1 and 2. We hope that this way the data will be more understandable to readers.
- the English language must be improved
The manuscript was checked by a native speaker and minor linguistic corrections were made.
Reviewer 2 Report
Comments and Suggestions for Authors The authors describe the knowledge and rate of recommended vaccinations in 300 patients treated in a specialized rheumatology outpatient clinic in the capital of Poland. The results are compared to 60 “controls”. Vaccination rates and knowledge about vaccinations are low among patients and controls.The work shows that there is a great need for optimization and the authors suggest 5 steps and an annual update.
The title should be worded more appellatively:Don’t leave your patients in the dark – using ACR and EULAR Recommendations
for vaccination in Polish adult patients with autoimmune inflammatory rheumatic diseases
The frequency of diagnoses, but especially of the medications administered and the recruitment to a specialized outpatient clinic in the capital, raise doubts about its transferability to the country of Poland. Data comparing vaccination rates in different parts of the country and population groups could be added here.
The most important figures should be mentioned in the abstract. Recruitment of the control group remains unclear and should be better explained. Different cohort sizes are confusing, sometimes 300, sometimes 280 for AIIRD and sometimes 60, sometimes 52 for controls. 2 patients with AIIRD have unknown diagnosis and unknown therapy. These patients should be deleted from the analysis.In Table 7, the lower proportion of the AIIRD patient data has slipped below the controls.
The work could be shortened somewhat if data presented in tables are not named identically in the text or vice versa Comments on the Quality of English Language
Minor changes could impove the readibility. Example the abstract:
Introduction: Patients with autoimmune inflammatory rheumatic diseases (AIIRD) have a higher risk of infection than the general population. According to ACR and EULAR recommendations, vaccination against influenza, COVID-19, pneumococcus and tetanus is recommended for most patients with AIIRD. OBJECTIVES: The study aimed to determine the vaccination coverage and factors influencing it in Polish AIIRD patients. Patients and Methods: The study was conducted at the National Institute of Geriatrics, Rheumatology and Rehabilitation in Warsaw, Poland between May 2023 and October 2023. Study participants received a questionnaire regarding knowledge of the recommendation, vaccination coverage, determinants influencing their decision to vaccinate, and their perspectives on immunization. Results: The study included 300 AIIRD patients and 60 controls. Comparably low vaccination coverage against all diseases was found in both groups (highest for COVID-19, lowest for pneumococcus). Patients in both groups had low knowledge of recommended vaccinations. AIIRD patients were also unaware that they should avoid live vaccines. The main motivations for vaccination among AIIRD patients were fear of infection and medical advice. Conversely, the predominant reasons for non-vaccination were lack of knowledge that vaccination is recommended and concerns about potential adverse effects (including exacerbation of rheumatic disease). Patients reported that they did not receive recommendations for vaccination from both primary care physicians and their rheumatologists. Conclusion: To improve vaccination coverage among AIIRD patients in Poland, we should educate AIIRD patients about vaccination during routine medical visits, emphasizing the increased risk of infection, informing them about recommended vaccinations, and clarifying doubts about adverse effects.
Author Response
Thank you for your insightful review of our work, which we hope allowed us to improve it.
- The title should be worded more appellatively: Don’t leave your patients in the dark – using ACR and EULAR Recommendations for vaccination in Polish adult patients with autoimmune inflammatory rheumatic diseases
Thank you for your suggestion, we have changed the title accordingly.
- The frequency of diagnoses, but especially of the medications administered and the recruitment to a specialized outpatient clinic in the capital, raise doubts about its transferability to the country of Poland. Data comparing vaccination rates in different parts of the country and population groups could be added here.
Thank you for this very interesting comment. The study was conducted in a reference center in the capital city, and one should be cautious about the generalization of its results to the rest of Poland, where it could be expected that vaccination rates and the level of knowledge about vaccinations are even worse. However, our data suggest this may not be the case. In the study, more than half of the patients were people from rural areas and small towns, and the place of residence (living in cities with over 500,000 inhabitants) only influenced the level of vaccination against COVID-19 (OR=3.78, 95%CI=1.67-8.52). Moreover, in the study, we compared the vaccination rate of people treated with bDMARDs/tsDMARDs (under the constant care of our reference center) with patients treated with cDMARDs (mainly patients sent from outside our center to start bDMARDs/tsDMARDs treatment) - finding no differences between the groups. We have updated the discussion with the issue raised.
- The most important figures should be mentioned in the abstract.
Thank you for your comment, we have updated the abstract with the most important figures.
- Recruitment of the control group remains unclear and should be better explained.
The control group consisted of patients hospitalized in the rheumatology department, referred for diagnostics - so far without a diagnosis of AIIRD, not under constant rheumatological care, and without immunomodulatory treatment. We have improved the Patients section to be clearer in this regard.
- Different cohort sizes are confusing, sometimes 300, sometimes 280 for AIIRD and sometimes 60, sometimes 52 for controls.
- In Table 7, the lower proportion of the AIIRD patient data has slipped below the controls.
Thank you for pointing out the errors in Table 7 that make it confusing. The number 280 referred to the number of patients under rheumatological care, and 52 to the number of control group patients under the care of a family doctor. We have corrected the Table 7 to make it clearer.
- 2 patients with AIIRD have unknown diagnosis and unknown therapy. These patients should be deleted from the analysis.
2 patients under the care of the Biologic Therapy Center did not complete their surveys with data enabling their surveys to be linked to patients' electronic medical records, making it impossible to determine their type of disease, disease duration, and exact type of treatment received. However, these surveys were complete in other respects, which made it possible to analyze the data collected in them.
- The work could be shortened somewhat if data presented in tables are not named identically in the text or vice versa
Thank you for this comment, we have edited the data in the Results section in such a way that it does not repeat the data from the tables.
- Minor changes could impove the readibility. Example the abstract
The manuscript was checked by a native speaker and minor linguistic corrections were made, including the abstract.
Reviewer 3 Report
Comments and Suggestions for Authors
The work presented by Jakub Wroński and collaborators is certainly interesting and of high scientific importance. The study presents vaccination rates in patients with pathologies of autoimmune origin and of rheumatological interest. The methods used are correct and the statistical analyzes are used correctly.
However, in the work there is a need for a paragraph on vaccination responses of immunocompetent patients who have responded or not to vaccination and also the response to the treatment of these type of patients with micophenolate
10.3390/v13112261 Robust and persistent b-and t-cell responses after covid-19 in immunocompetent and solid organ transplant recipient patients.
10.3390/v14081766 Ongoing Mycophenolate Treatment Impairs Anti-SARS-CoV-2 Vaccination Response in Patients Affected by Chronic Inflammatory Autoimmune Diseases or Liver Transplantation Recipients: Results of the RIVALSA Prospective Cohort
It is suggested to implement this chapter in the introduction as it is necessary to give greater prominence to the paper presented.
After these small revisions I strongly suggest publication
Author Response
- The work presented by Jakub Wroński and collaborators is certainly interesting and of high scientific importance. The study presents vaccination rates in patients with pathologies of autoimmune origin and of rheumatological interest. The methods used are correct and the statistical analyzes are used correctly.
- After these small revisions I strongly suggest publication
Thank you for your kind review of our work.
- However, in the work there is a need for a paragraph on vaccination responses of immunocompetent patients who have responded or not to vaccination and also the response to the treatment of these type of patients with micophenolate 10.3390/v13112261 Robust and persistent b-and t-cell responses after covid-19 in immunocompetent and solid organ transplant recipient patients. 10.3390/v14081766 Ongoing Mycophenolate Treatment Impairs Anti-SARS-CoV-2 Vaccination Response in Patients Affected by Chronic Inflammatory Autoimmune Diseases or Liver Transplantation Recipients: Results of the RIVALSA Prospective Cohort It is suggested to implement this chapter in the introduction as it is necessary to give greater prominence to the paper presented.
Thank you for this valuable comment. Indeed, immunomodulatory drugs, especially rituximab, mycophenolate mofetil, abatacept, and glucocorticosteroids, may reduce the immunogenicity of vaccinations in AIIRD patients and this is a factor that should be taken into account when planning infection prevention in this group. Although this is not the topic of our work, we agree that including this information in the introduction may give a broader context to our work. We have updated the introduction section (due to the extensiveness of literature data dealing with this issue we cited the literature review we performed in our other recent article).